# Extending RML to further embrace data source heterogeneity

Pano Maria[1,a]

[1]Skemu, Rotterdam, The Netherlands
[a]pano@skemu.com

**Abstract.** The RDF Mapping Language (RML) has provided a language for specifying transformation of heterogeneous structured data sources to RDF. RML was extended from R2RML, a W3C recommendation for transforming relational data to RDF, but in applying RML in practice we've identified several aspects where RML is still more table oriented, than oriented towards data format heterogeneity, limiting its potential.

In this position paper, these aspects are presented and discussed, and possible extensions to RML are proposed. In doing so, we hope to increase the usability of RML for transforming heterogeneous data sources to RDF.

**Keywords:** RML · RML mapper · RDF · RDF generation · R2RML · ETL

## 1 Introduction

During the past two years of having employed RML in ETL processes in a production setting, transforming CSV, XML, and JSON sources to RDF, and having developed an RML engine[1], some aspects of the current RML specification[2] that limit its potential application have come to light. The existence of these aspects can be attributed to the fact that RML was extended from R2RML [1], which was designed for mapping relational data to RDF, in which, due to the tabular nature of relational data, these aspects dit not form an issue. It seems that when RML was created, these issues were not identified. Hopefully, our experience of using RML on a variety of data sources in a production setting can be used to improve these aspects of RML.

In this position paper, we present and discuss the aforementioned issues and propose solutions to overcome them. The aim of this paper is to raise awareness for these issues in the current RML specification and to start a process of improving these parts of the RML specification.

[1] https://github.com/carml/carml
[2] http://rml.io/spec.html

## 2 Support for, or clarification of, multiplicity of term maps

R2RML provided the possibility of transforming data from relational databases to RDF using a declarative mapping language. RML has expanded R2RML, providing the capability to transform any queryable structured data source to RDF [2]. Naturally, this newfound flexibility brings with it some new challenges.

Some of these challenges stem from the fact that the current way of processing sources with RML is still optimized for tabular data, whereas XML and JSON data sources are hierarchically structured. One of the issues that arises with hierarchically structured data sources, when applying RML, is that of the multiplicity of term maps.

In R2RML, it was not necessary to specify multiplicity of term maps, since multiplicity in tabular data is expressed by repeating values over different rows (Table 1). With R2RML, triples are mapped in iterations over the source and the table row is the unit of iteration. Hence, multiplicity is captured automatically.

Table 1. Multiplicity expressed in tabular data.

| NAME | CAR |
| --- | --- |
| John Doe | BMW |
| John Doe | Seat |
| John Doe | Porsche |
| Richard Roe | Toyota |
| Richard Roe | Tesla |

In hierarchically structured data sources, this is usually not the case. In these data sources, multiplicity is expressed within nodes, e.g. using repeating sub-nodes in XML, or an array in JSON.

```
1   {
2     "cardata" : [{
3       "name":"John Doe",
4       "cars":[ "BMW", "Seat", "Porsche" ]
5     }, {
6       "name":"Richard Roe",
7       "cars":[ "Toyota", "Tesla" ]
8     }]
9   }
```

Listing 1. Multiplicity in JSON

Say we want to express which cars a person has, based on Listing 1, using an RML mapping. For this we would need a way to get all the values from the `cars` arrays nested in the `cardata` objects.

So how does this work with RML? The RML specification is, in fact, unclear on whether it supports generating multiple terms from term maps or not. It is not mentioned in the specification, and the R2RML definition (Listing 2) of a term map is not extended in RML, nor are any new constructs introduced that handle the generation of multiple terms. However, some examples seem to imply multiplicity is, or should be, supported.[3]

```
1   :TermMap rdf:type owl:Class ;
2     rdfs:label "Term Map"@en ;
3     rdfs:comment "A function that generates an RDF term from a logical
        table row."@en .
```

Listing 2. The R2RML definition of term map [1]

Since RML is used on hierarchically structured data sources, and term generating functions will inevitably lead to multiple results, it is necessary to either clarify this aspect in the RML specification, or introduce new constructs which will support the generation of multiple RDF terms.

To this end, we propose introducing new RDFS super classes and RDFS super properties for R2RML mapping constructs, to explicitly define support for multiple values. Defining these constructs using RDFS allows for maintaining backwards compatibility for RML mappings that use the R2RML constructs, either through RDFS inferencing [3], or support by RML engines.

We further propose that the concrete term map types `:SubjectMap`, `:PredicateMap`, `:ObjectMap`, `:RefObjectMap`, and `:GraphMap` could all have multiple terms as a result. Thus the execution of a triples map could lead to triples for:
• multiple subjects, one for each item in the set of subject map function results
• with, for each subject,
  ◦ for each predicate object map

- predicates for each item in the set of predicate map function results, and
- objects for each item in the set of (referencing) object map function results,

  ∘ and a graph assignment for each item in the set of graph map results.

For referencing object maps, the multiplicity would be dependent on the specified join. If a join condition's child expression results in multiple values, and one of these values matches with a result from its parent expression, this would constitute a valid join, meaning that a referencing object map may generate as many terms as can be validly joined with it.

As an example, applying this proposal and using the data from Listing 1 as the source we can create a mapping (Listing 3) which references the cars property in the car data objects.

```
1   :CarDataMapping a rr:TriplesMap ;
2     rml:logicalSource [
3       rml:source "cardata.json" ;
4       rml:referenceFormulation ql:JSONPath ;
5       rml:iterator "$.cardata" ;
6     ] ;
7
8     rr:subjectMap [
9       rr:template "http://www.example.com/{name}" ;
10    ] ;
11
12    rr:predicateObjectMap [
13      rr:predicate ex:ownsCar ;
14      rr:objectMap [
15        rml:reference "cars" ;
16      ] ;
17    ] ;
18  .
```

Listing 3. Example mapping with reference to multiple values

Applying this mapping to the data would result in the RDF data in Listing 4.

```
1   ex:John%20Doe
2     ex:ownsCar
3       "BMW" ,
4       "Seat" ,
5       "Porsche" .
6   ex:Richard%20Roe
7     ex:ownsCar
8       "Toyota" ,
9       "Tesla" .
```

Listing 4. RDF car data resulting from the RML mapping of Listing 3

Adding to this example a predicate object map with a referencing object map containing a join with multiple values (Listing 5)

```
1    rr:predicateObjectMap [
2      rr:predicate ex:ownsCar2 ;
3      rr:objectMap [
4        rr:parentTriplesMap :CarMapping ;
5        rml:joinCondition [
6          rr:child "cars" ;
7          rr:parent "$" ;
8        ] ;
9      ] ;
10   ] ;
11 .
12
13 :CarMapping a rr:TriplesMap ;
14 rml:logicalSource [
15   rml:source "cardata.json" ;
16   rml:referenceFormulation ql:JSONPath ;
17   rml:iterator "$.cardata[*].cars" ;
18 ] ;
19
20 rr:subjectMap [
21   rr:template "http://www.example.com/{$}" ;
22 ] ;
23
24 rr:predicateObjectMap [
25   rr:predicate rdfs:label ;
26   rr:objectMap [
27     rml:reference "$" ;
28     rr:datatype xsd:string ;
29   ] ;
30 ] ;
31 .
```

Listing 5. Example mapping joining multiple values

we get the result in Listing 6.

```
1  ex:John%20Doe
2    ex:ownsCar2
3      ex:BMW ,
4      ex:Seat ,
5      ex:Porsche .
6
7  ex:BMW rdfs:label "BMW" .
8
9  ex:Seat rdfs:label "Seat" .
10
11 ex:Porsche rdfs:label "Porsche" .
12
13 ex:Richard%20Roe
14   ex:ownsCar2
15     ex:Toyota ,
16     ex:Tesla .
17
18 ex:Toyota rdfs:label "Toyota" .
19
20 ex:Tesla rdfs:label "Tesla" .
```

Listing 6. RDF car data resulting from the RML mapping of listing 5

Although this example is overly simplified, it shows the necessity of support for generating multiple terms with RML mapping constructs.

[3] http://rml.io/RMLmappingLanguage.html

# 3 Specifying data sources

RML has enabled us to transform various data sources to RDF. However, in working with RML, one can only conclude that the possibilities of describing the data sources to be transformed are limited. To describe a data source, the specification only provides a property `rml:referenceFormulation` with which to express the data format of the source, and the `rml:source` property, with which to point out a source's location.

It is often useful to be able to specify more information about a certain source in order to facilitate the mapping process. Since the range for `rml:source` is undefined (Listing 7), it can easily be used to point to more descriptive source representations.

```
1   rml:source rdf:type rdf:Property ;
2       rdfs:label   "source" ;
3       rdfs:comment "qualified name of the source data."@en ;
4       rdfs:domain  rml:LogicalSource
5     .
```

Listing 7. Definition of `rml:source` in the RML ontology.

To describe our data sources, we can define a class `rml:Source`. This class represents an abstract data source. In addition, we define a sub-class `rml:FileSource`, which represents those data sources that are physical files, and can have a property `rml:url` to specify their respective location (Listing 8).

```
1   rml:Source a owl:Class ;
2     rdfs:label "Source"@en ;
3     rdfs:comment "A source that can be mapped to RDF."@en ;
4   .
5
6   rml:FileSource a owl:Class ;
7     rdfs:label "FileSource"@en ;
8     rdfs:subClassOf rml:Source ;
9     rdfs:comment "A file based source."@en ;
10  .
11
12  rml:url a rdf:Property ;
13    rdfs:label "url"@en ;
14    rdfs:domain rml:FileSource ;
15    rdfs:comment "A value referencing the location of the FileSource"@en
16  .
```

Listing 8. Definition of an abstract source and a file source contruct.

These constructs do not yet add any new functionality to RML, but they provide an extension point for several useful descriptors of a data source. In the following sub-sections, we will explore some of these descriptors.

## 3.1 Support for input streams

The absence of source descriptors, other than a location, makes it difficult to provide useful information about the data source to RML engines. A prominent use case for RML is to be used as part of an ETL process, where an RML engine is fed a stream of source data to be transformed according to an RML mapping. In this case, there is no specific file, nor file location the engine should be concerned with. In fact, the engine will most likely expect one or more input streams to be provided, upon which it can act. Yet, there is no way of describing an input stream using RML.

To solve this, we propose extending the `rml:Source` defined earlier with a sub-class `rml:Stream` to represent those sources that are streams of data (Listing 9).

```
1   rml:Stream a owl:Class ;
2     rdfs:label "Stream"@en ;
3     rdfs:subClassOf rml:Source ;
4     rdfs:comment "A stream is a sequence of data."@en ;
5   .
6
7   rml:streamName a owl:DatatypeProperty ;
8     rdfs:label "stream name"@en ;
9     rdfs:domain rml:Stream ;
10    rdfs:range xsd:string ;
11    rdfs:comment "A name to identify a stream."@en ;
12  .
```

Listing 9. Definition of stream data source.

In order to distinguish between multiple streams, a `rml:streamName` property is defined as well. An RML engine can use these source descriptors to identify the correct streams of data corresponding with each triples map.

### 3.2 Specifying source encoding

As RML opens up support of transforming any structured source to RDF, so comes with it the challenges of interpreting strings in different encodings. Currently missing from RML is a way to specify the encoding of data sources. As a result, data source encoding must either be specified out of band, or an engine must employ its own strategies to handle encoding.

A proposed solution to this is the addition of `rml:encoding` as property of `rml:Source` (Listing 10).

```
1   rml:encoding a rdf:Property ;
2     rdfs:label "encoding"@en ;
3     rdfs:domain rml:Source ;
4     rdfs:comment "A property that specifies the encoding source."@en ;
5   .
```

Listing 10. Definition of encoding property.

Using this newly defined property, we can describe the encoding of a specific data source using a controlled list of known encodings[4] (Listing 11).

```
1   ex:LS a rml:LogicalSource ;
2     rml:source [
3       a rml:FileSource ;
4       rml:url "/location/of/file.csv" ;
5       rml:encoding "utf-16le" ;
6     ] ;
7     rml:referenceFormulation ql:CSV ;
8   .
```

Listing 11. Example of a file source with specified encoding.

The introduction of this property affords an RML engine to use the specified encoding to correctly read and transform the source document to RDF.

[4] https://encoding.spec.whatwg.org/

### 3.3 Support for namespaces declarations for XML sources

When working with XML data sources and RML, the query language of choice is XPath [2]. It is common practice to use namespaces in XML documents as well as prefixes to shorten the namespaced node names. Unfortunately, XPath has no standard way of specifying namespaces to be used in queries. Thus a query that one would like to write as `//top10nl:FeatureMember/top10nl:Wegdeel`, must instead be written `//[local-name()='FeatureMember'` and `namespace-uri()='http://example.org/xmlns/top10nl/']/` `[local-name()='Wegdeel' and namespace-uri()=` `'http://example.org/xmlns/top10nl/']`.

Luckily, most XPath implementations support the registration of namespaces in order to keep the queries legible and maintainable[5, 6]. However, currently there is no way to specify the use of namespaces in XPath queries used in an RML mapping in order to leverage this functionality.

An extension of `rml:Source` is, therefore, proposed to describe this aspect of XML data sources (Listing 12).

```
1   rml:XmlSource a owl:Class ;
2     rdfs:label "XmlSource"@en ;
3     rdfs:subClassOf rml:Source ;
4     rdfs:comment "An XML source"@en ;
5   .
6
7   rml:declaresNamespace a owl:ObjectProperty ;
8     rdfs:label "declaresNamespace"@en ;
9     rdfs:domain rml:XmlSource ;
10    rdfs:range rml:Namespace ;
11    rdfs:comment "A namespace declaration that is used to support
    namespaces in XML document references."@en ;
12  .
13
14  rml:Namespace a owl:Class ;
15    rdfs:label "Namespace"@en ;
16    rdfs:comment "A document namespace."@en ;
17  .
18
19  rml:namespacePrefix a owl:DatatypeProperty ;
20    rdfs:label "namespacePrefix"@en ;
21    rdfs:domain rml:Namespace ;
22    rdfs:range xsd:string ;
23    rdfs:comment "The prefix value of a namespace."@en ;
24  .
25
26  rml:namespaceName a owl:DatatypeProperty ;
27    rdfs:label "namespaceName"@en ;
28    rdfs:domain rml:Namespace ;
29    rdfs:range xsd:string ;
30    rdfs:comment "The name value of a namespace."@en ;
31  .
```

Listing 12. Definition of XML source and namespace constructs.

We define a class `rml:XmlSource` which can have a property `rml:declaresNamespace`, which has as its range the class `rml:Namespace`. This class has properties `rml:namespacePrefix` and `rml:namespaceName` to specify the prefix and the name of a namespace of the source XML respectively.

Note that one can combine `rml:XmlSource` with either `rml:FileSource` or `rml:Stream` to specify its input form (Listing 13).

```
1   ex:LS a rml:LogicalSource ;
2     rml:source [
3       a rml:Stream, rml:XMLSource ;
4       rml:streamName "ex-stream" ;
5       rml:declaresNamespace [
6         rml:namespacePrefix "top10nl" ;
7         rml:namespaceName "http://example.org/xmlns/top10nl/" ;
8       ] ;
9     ] ;
10  .
```

Listing 13. Example of an XML stream source with namespace declaration.

This proposed addition makes it possible to leverage the namespace support of XPath query libraries, enabling the use of prefixes in XPath expressions in RML mapping constructs.

[5] https://www.saxonica.com/html/documentation/javadoc/net/sf/saxon/s9a-pi/XPathCompiler.html#declareNamespace-java.lang.String-java.lang.String-

[6] https://www.npmjs.com/package/xpath#default-namespace-with-mapping

## 4   Conclusions

RML is a seminal development in the field of RDF generation, facilitating the transformation of a variety of data sources. However, there are still some improvements that can be made to the RML specification to further embrace the diverse nature of the data sources to be transformed.

Overall, we observe that multiplicity of term maps is currently not clearly defined in the RML specification, making it difficult for RML engines to be correctly implemented. We propose extending the RML vocabulary and specifying constructs that clearly support the generation of multiple RDF terms.

RML aims to provide support for heterogeneous data sources, yet is limited in its vocabulary to describe these data sources. We propose extensions to RML that allow for more descriptive data source specifications, thereby improving the user experience of creating mappings, as well as allowing RML to correctly handle the variety of data sources.

The aim of this position paper is to start a discussion around the aforementioned issues. Our hope is that this discussion will lead to extensions of RML that mitigate these issues, further improving RML's usability and applicability in practice.

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
