# OpenReview forum: "Extending RML to further embrace data source heterogeneity"
_eswc-conferences.org/ESWC/2019/Workshop/KGB — Submitted to KGB 2019_

### Official Review · ~Franck_Michel1 · 2019-03-26
**Pragmatic but very modest contribution**

**Rating:** 2
**Confidence:** 3

**Review:**

This short article proposes extensions to the RML mapping language, inspired by the experience of the author in using RML in real-life use cases. The extensions pertain to the generation of multiple RDF terms in the case of structured documents (typically JSON arrays), and the description of alternate types of data sources.

Overall this is a very modest contribution, merely reporting on extensions implemented by the author to address specific mappings cases. If the cases are probably of interest to a larger community of users, the article remains focused on them and does not try to generalize and address a wider scope of cases. The justifications of the proposed changes in section 2 are definitely not convincing, assuming that the feature "probably" lacks in RML, but with no certainty. Besides, there is no mention of any related works. In particular, the proposition in section 2 is already fully covered by xR2RML [1,2] with a more generic approach.

Consequently, although the article would succeed in opening the discussion with the RML community, which is the goal, it hardly reaches the quality level that is expected, even for a conference workshop. Hence the mark 2.

Below I describe my concerns in more details for sections 2 and 3.

§2.
The discussion about multiple terms of term maps lacks convincing motivation. You write: "it  is  necessary  to  either clarify this aspect in the RML specification, or introduce new constructs which will support the generation of multiple RDF terms".
Then, you opt for the second option. Well, the least would be to check option 1 with RML's authors. If this is just a matter of updating the specification, then your proposition is useless. You also write: "some examples seem  to imply multiplicity is, or should be, supported.". You cannot be satisfied with such a vague assessment, you should make sure of that.

Furthermore, the model you propose to support term maps generating multiple terms is no different from what xR2RML proposed [1]. See in particular the detailed discussions, in the language specification [2], on the various iteration models (§3.1.3) (xrr:nestedTermMap) and on the production of multiple terms (§3.2.2.3).
You could also consider other issues like the fact that, within an iteration (rml:iteartion), it is not possible to access fields of the document that are above the current iteration level. xR2RML deals with such cases with the xrr:pushDown predicate that pushes fields from a given level in the document hierarchy, into sub-documents [2].

Your proposition in listing 3 totally ignores the expressiveness of JSONPath. The problem is that, according to JSONPath, expressions "$.cardata" and "$.cars" denote arrays. Why not leverage JSONPath and write "$.cardata.*" and "$.cars.*" instead, as in xR2RML, this states more clearly that the referenced element is not an array but each and every element within the array.


§3.
You define a new class rml:Source although there already exists an rml:LogicalSource class (http://rml.io/ns/rml), that is not described further though. The proposed rml:FileSource should better be a subclass of rml:LogicalSource.

--

[1] Michel F., Djimenou L., Faron-Zucker C. & Montagnat J. (2015). Translation of Relational and Non-Relational Databases into RDF with xR2RML. In Proceeding of the 11th international conference on Web Information Systems and Technologies (WebIST), pp. 443–454. Lisbon, Portugal.

[2] Michel F., Djimenou L., Faron-Zucker C. & Montagnat J. (2014). xR2RML: Non-Relational Databases to RDF Mapping Language. Report ISRN I3S/RR 2014-04-FR. <http://hal.archives-ouvertes.fr/hal-01066663>

---

### Official Review · ~Freddy_Priyatna1 · 2019-03-28
**interesting yet immature paper that may trigger a relevant workshop discussion**

**Rating:** 3
**Confidence:** 3

**Review:**

This position paper proposes a set of new classes/properties as an extension of RML, driven that the author´s experience of applying RML in practice. The paper is easy to read and the author has done a good job providing an example for each part of the proposal. However, there are a couple of missing information here that should be provided. First is the setting and context of the author's experience, what type of industry/datasets format/dataset size, etc. Second is comparison with other existing proposals such as xR2RML or kR2RML for multiple or nested term maps  and morph-stream for including window operator in mappings.

Considering that this is a position paper yet a very relevant one to the workshop, I would like to see this paper presented which I believe may trigger a discussion on how should RML being extended to support (further) heterogeneity aspect.

---

### Official Review · AnonReviewer3 · 2019-04-01
**These days, more may be required, even from a workshop paper**

**Rating:** 2
**Confidence:** 2

**Review:**

This is a welcome but rather immature paper. The problem studied (i.e., revisiting RML in the case of various kinds non-relational data sources such as XML and JSON) is certainly interesting for the workshop participants and may generate discussion I think.

However, the paper needs more work to be persuasive starting with discussing more related work e.g.,
http://events.linkeddata.org/ldow2018/papers/LDOW2018_paper_11.pdf

Related work is generally missing from the paper, and therefore the paper fails to convince the reader about the contributions of the work  with respect to the state of the art.

I suggest rejection.

---

### Decision · Program_Chairs · 2019-04-08
**Acceptance Decision**

**Decision:**

Reject

**Comment:**

This contribution is rejected for inclusion in the proceedings.